# Improving access to specialist care in correctional facilities through Ontario eConsult

**Danica Goulet**[1,2], **Claire Sethuram**[1], **Erin Keely**[2,3,4], **Clare Liddy**[1,2,5,6,7] *

**1** C. T. Lamont Primary Healthcare Research Centre, Bruyère Health Research Institute, Ottawa, Ontario, Canada, **2** (Ontario) eConsult Centre of Excellence, The Ottawa Hospital, Ottawa, Ontario, Canada, **3** Division of Endocrinology and Metabolism, The Ottawa Hospital, Ottawa, Ontario, Canada, **4** Department of Medicine, University of Ottawa, Ottawa, Ontario, Canada, **5** Department of Family Medicine, Faculty of Medicine, University of Ottawa, Ottawa, Ontario, Canada, **6** Ottawa Hospital Research Institute, Ottawa, Ontario, Canada, **7** Ontario Health, Toronto, Ontario, Canada

* cliddy@uottawa.ca

**Data Availability Statement:** A minimal dataset cannot be provided due to the potential presence of identifiable information and because the data is owned by a third-party organization (Ontario

## Abstract

### Objectives

To evaluate the accessibility of multispecialty advice for primary care providers (PCPs) within correctional facilities, catering to the healthcare needs of individuals in federal custody in Ontario, Canada, through the utilization of electronic consultation (eConsult).

### Design

Retrospective, cross-sectional, descriptive analysis.

### Setting

eConsults submitted by PCPs within federal correctional facilities through the Ontario eConsult Service between April 1st, 2019, and March 31st, 2023.

### Participants

906 completed eConsults were submitted by 21 PCPs in correctional facilities.

### Results

The top three specialties sent to were cardiology (46%, N = 417), dermatology (14%, N = 128), and endocrinology and metabolism (8%, N = 68). The median specialist response time was 0.9 days. The median time specialists spent responding to each case was 15 minutes. PCPs received advice on a new or additional course of action in 34% of eConsult cases. In-person specialist appointments were avoided in 81% of cases.

### Conclusions

Ontario eConsult provides an ideal venue to improve access to multispecialty advice for people who are incarcerated. This service reduces the need for face-to-face specialist visits,

Health). The data used in this study is provided by Ontario Health (OTN Business Unit) (OH (OTN)) through a Data Sharing Agreement with the Ontario eConsult Centre of Excellence (COE), the lead organization for the Ontario eConsult program. OH (OTN) supplies the COE with transactional and record-level data generated through the use of the Ontario eConsult service. This data is used to generate reports on the utilization of the program, facilitating troubleshooting and supporting the ongoing development of the eConsult program. The authors have obtained permission to use the data set through a formal Data Sharing Agreement between Ontario Health (OTN Business Unit) and the Ontario eConsult Centre of Excellence, the lead organization for the Ontario eConsult program. This agreement ensures compliance with privacy regulations and outlines the terms of data access and usage. The authors did not receive any special privileges in accessing the data beyond the terms outlined in the Data Sharing Agreement. All data access and usage were conducted in accordance with the agreement, ensuring that the same terms apply to other researchers seeking access to the data. To apply for access to the data, individuals can contact the Ontario Health (OTN Business Unit) Privacy Office through the following methods: • Phone: 1-855-654-0888 • Email: privacy@otn.ca Additionally, requests for de-identified aggregate Ontario eConsult utilization data can be submitted through the following online form: https://form. asana.com/?k=zmEBJuRNGb9_YLGOXyqYLg&d= 423266023889752 Data requests can be made through the following form: https://form.asana. com/?k=zmEBJuRNGb9_YLGOXyqYLg&d= 423266023889752.

**Funding:** "Funding for this project was provided by the Ontario Ministry of Health and Long-Term Care (MOHLTC). The opinions, results, and conclusions reported in this paper are those of the authors and are independent of the funding sources. No endorsement by the Ontario MOHLTC is intended or should be inferred. The funders had no role in study design, data collection and analysis, decision to publish, or preparation of the manuscript".

**Competing interests:** EK is the Executive Director and CL is the Evaluation Lead of the Ontario eConsult Centre of Excellence, which is responsible for the Ontario eConsult program, with both receiving salary support from the Ontario Ministry of Health. EK answers occasional eConsults (less than 1 per month) as a specialist through the service, for which she is reimbursed. This does not alter our adherence to PLOS ONE policies on sharing data and materials."

decreases cost-of-care, and avoids unnecessary transportation outside of correctional facilities with potential security issues.

## Introduction

Every year, there are over 80,000 adult admissions to correctional facilities in Ontario, Canada [1], and on an average day, more than 10,000 persons are held in custody [2, 3]. Of those held in custody, approximately 68% serve sentences of less than two years under provincial/territorial custody, while the remaining 32% serve sentences of over two years under federal custody [4]. Compared to the general population, persons in custody within Canadian correctional facilities experience higher rates of mortality, suicide, and chronic health conditions, as well as increased utilization of ambulatory care, emergency departments, medical-surgical hospitalization, and psychiatric hospitalization [5, 6]. They also face significantly lower life expectancies and higher rates of latent tuberculosis, sexually transmitted infections, hepatitis C, HIV, and other blood-borne infections than the general population [5, 7]. Many of these diseases require specialist care; however, people who are incarcerated face significant challenges accessing it, including transportation to in-person appointments and the requirement for a minimum of two prison officers to accompany each incarcerated individual outside of the correctional facility [5, 8]. This not only results in additional costs for the department of corrections, but also leads to logistical challenges such as transfers between institutions, ultimately causing missed appointments and long wait times. Improving access to care for incarcerated individual is crucial, as it can significantly enhance their health outcomes and reduce the burden of disease.

In June 2018, an approach to improve access to care was implemented in Ontario, which allows primary care providers (PCPs) to send questions to specialists regarding a patient's care on a secure web-based application through the Ontario eConsult Service, facilitating asynchronous communication between clinicians [9]. Specialists respond in one day on average, and two-thirds of cases are resolved without the patient needing a face-to-face specialist appointment [10]. eConsult has the potential to address issues with access to care across a range of health system contexts including correctional settings [11–13].

A scoping review by Sethuram *et al.* (2022) identified 13 unique electronic consultation systems serving incarcerated individuals within correctional facilities across six countries, including Canada, with evidence that the use of electronic consultation in this setting is feasible, beneficial, cost-effective, increases access to care, has a positive impact on clinical care, and avoids unnecessary transportation of incarcerated individuals outside of the facilities and potential security issues [14]. While eConsult is currently operating in many correctional facilities worldwide, it is worth noting that more than half of them (7 out of the 13) provide access to a single specialty group exclusively, with dermatology and ophthalmology being the most common [14]. This contrasts with the Ontario eConsult Service described here, where PCPs in correctional facilities have access to over 120 specialties offered province-wide.

In this study, we evaluated the utilization patterns and accessibility of multispecialty advice through Ontario eConsult for PCPs in correctional facilities and measured the impact of eConsult on the care of incarcerated populations across the province. The volume of eConsults sent, the range of accessed specialties, and the impact of the eConsult on the need for face-to-face referrals were examined to demonstrate the benefit of eConsult and serve as a template for broader implementation across Canada and in other jurisdictions.

## Methods

### Design

We conducted a retrospective, cross-sectional descriptive analysis of eConsults sent by PCPs in federal correctional facilities across Ontario, Canada. This study was submitted to the Ottawa Health Science Network Research Ethics Board (OHSN-REB) as a Quality Improvement study. The OHSN-REB waived its review, thus exempting the project from requiring full ethics approval. The ethics committee determined that this study falls under program evaluation and waived the requirement for informed consent for this study as it is a secondary data analysis.

### Setting

The Ontario eConsult Service operates within Canada's most densely populated province, home to over 14 million individuals [15], representing nearly 40% of the entire Canadian population, and served by approximately 15,000 family physicians [16] and 5,000 nurse practitioners [17]. Identical to other provinces in Canada, Ontario administers a Medicare program that is federally funded but managed at the provincial level and offers healthcare services to residents without any cost. Ontario Health is connected to communities and provider partners through six regions: West, Central, Toronto, East, North East, and North West. These regions serve as administrative divisions to facilitate the coordination and delivery of healthcare services across the province.

The Ontario eConsult Service is accessible to all PCPs in Ontario, including those working in institutionalized settings. In Canada, the administration of correctional services is a shared responsibility between the federal and the provincial/territorial governments [4]. The Correctional Service of Canada is responsible for the federal system and has jurisdiction over adults who are incarcerated serving custodial sentences of two or more years, whereas adults serving custodial sentences that are less than two years, or who are being held while awaiting trial or sentencing, fall under the authority of the provincial/territorial correctional service programs [4]. The province of Ontario is responsible for 8 federal institutions, seven for men and one for women, where the majority (6 out of 8) are located in the Ontario Health East region.

It is also important to note the Coronavirus-19 (COVID-19) outbreak emerged during our study period, where the World Health Organization (WHO) declared it a pandemic on March 11, 2020, recognizing its widespread and severe impact on public health and economies worldwide.

### The Ontario eConsult Service

The Ontario eConsult Service was implemented in Ontario in June 2018, enabling PCPs to communicate with specialists securely and asynchronously via an electronic platform. This service allows PCPs to consult specialists with clinical questions about their patients, providing timely advice on patient management and often eliminating the need for in-person specialist visits. The PCP submitting the eConsult has access to specialists in two models, through a managed specialty group model, where the case is assigned to a specialist from the chosen group, or direct-to-specialist (DTS). The Ontario eConsult Service is funded by the province's Ministry of Health and is provided to PCPs across Ontario at no charge. In fact, PCPs are remunerated at a flat rate per case, and specialists are remunerated at an hourly rate prorated to their self-reported billing time.

### Data collection

eConsult utilization data from the Ontario eConsult Service is routinely collected from the service. At the end of each eConsult case, PCPs complete a mandatory close-out survey to

evaluate (1) whether the specialist response confirmed their original course of action or provided them with new or additional information, and (2) the eConsult's impact on the need for a face-to-face referral.

### Data analysis

Descriptive statistics were used to analyze the last four years of Ontario eConsult data (April 2019 to March 2023). Time stamps for each eConsult communication were assessed to examine the response interval for specialist responses, and self-reported time billed was used to determine the amount of time each specialist spent responding to the eConsult.

## Results

During the study period, 906 completed eConsults were submitted by PCPs in federal correctional facilities. A total of 21 PCPs submitted at least 1 eConsult, with 12 being family physicians and 9 nurse practitioners. All 21 submitting PCPs practice in correctional facilities in the Ontario Health East region. Of the 21 PCPs, 67% submitted 10 or more eConsults and 33% sent more than 50 eConsults during the study period. In addition, 44% of eConsults (N = 397) were submitted by delegates (i.e., administrative support) on behalf of the PCP.

The number of eConsults sent increased more than 2-fold from the initial 12 months to the last 12 months of the study period (Fig 1).

PCPs were submitted directly to a specialist, through the DTS model (62%, N = 563), more frequently than to a managed specialty group (38%, N = 343). Of the 30 specialties accessed by PCPs, the top three specialties sent to were cardiology (46%, N = 417), dermatology (14%, N = 128), and endocrinology and metabolism (8%, N = 68). The most frequently accessed specialties are outlined in Fig 2.

A total of 178 specialists answered at least 1 eConsult from the 906 completed eConsult cases during the study period. The median specialist self-reported billing time was 15 minutes per case (interquartile range (IQR): 10). Of the 906 completed eConsults, the median response time was 0.9 days (IQR: 1.8) with 95% answered within 7 days, and 100% of cases answered within 30 days or less.

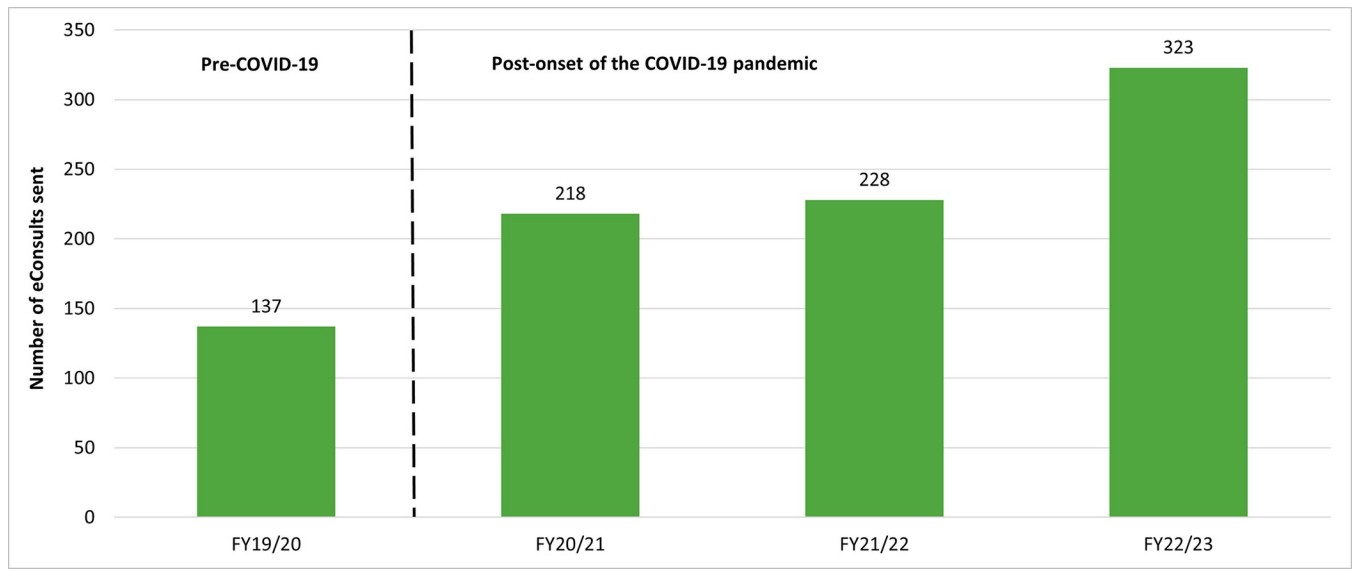

**Fig 1. Volume of eConsults sent by fiscal year (FY) (N = 906).**

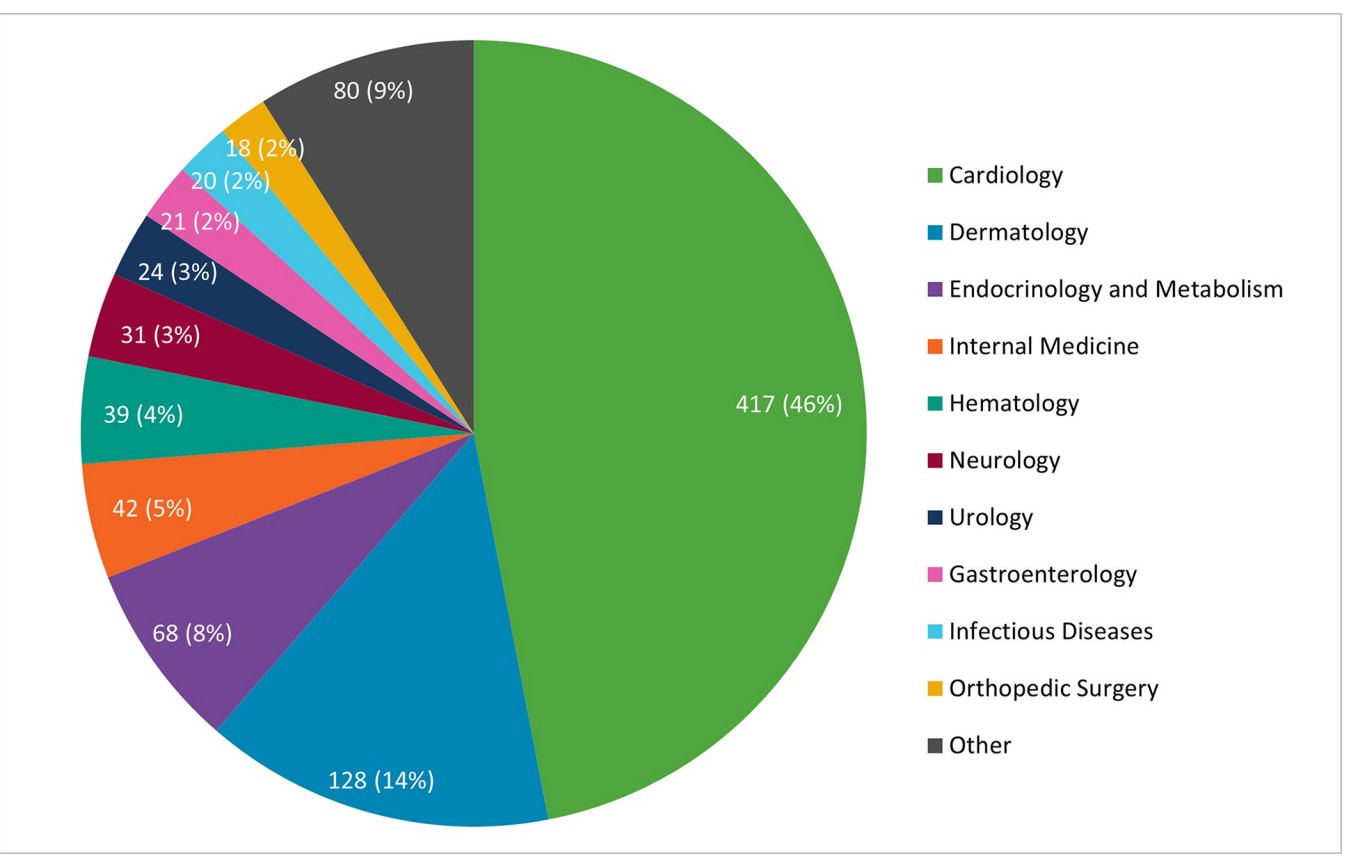

**Fig 2. Speciality distribution of eConsults sent (N = 906).**

PCPs completed the close-out survey for 890 of the 906 closed cases. Survey results indicated PCPs were able to confirm a course of action they originally had in mind in 580 cases (65%) and received advice on a new or additional course of action in 304 cases (34%) (Fig 3). In addition, a referral for a face-to-face specialist visit was first contemplated but now avoided as a result of the eConsult in 485 cases (54%) and was still not needed in 27% of cases (N = 237) (Fig 4). Alternatively, PCPs stated they initially contemplated a referral and still found it necessary after the eConsult in 143 cases (16%) (Fig 4).

## Discussion

Our study demonstrates the effective provision of prompt, multispecialty advice for PCPs within correctional facilities, catering to the healthcare needs of individuals in federal custody via the Ontario eConsult Service. Contrary to the majority of eConsult solutions serving incarcerated individuals worldwide [14], the Ontario eConsult Service provides PCPs access to over 120 specialties province-wide.

Cardiology emerged as the most frequently accessed specialty in our study, accounting for 46% of all eConsults from correctional facilities. This contrasts dramatically with the most frequently accessed specialty by the general public at the provincial level during the same time period—dermatology—which accounted for 15% of all eConsults, whereas cardiology, which ranked at number 8, accounted for 4% of all eConsults. The high demand for access to cardiology in our study is believed to be due to PCPs in correctional facilities seeking cardiologists' guidance regarding cardiac testing for people who are incarcerated undergoing opioid agonist

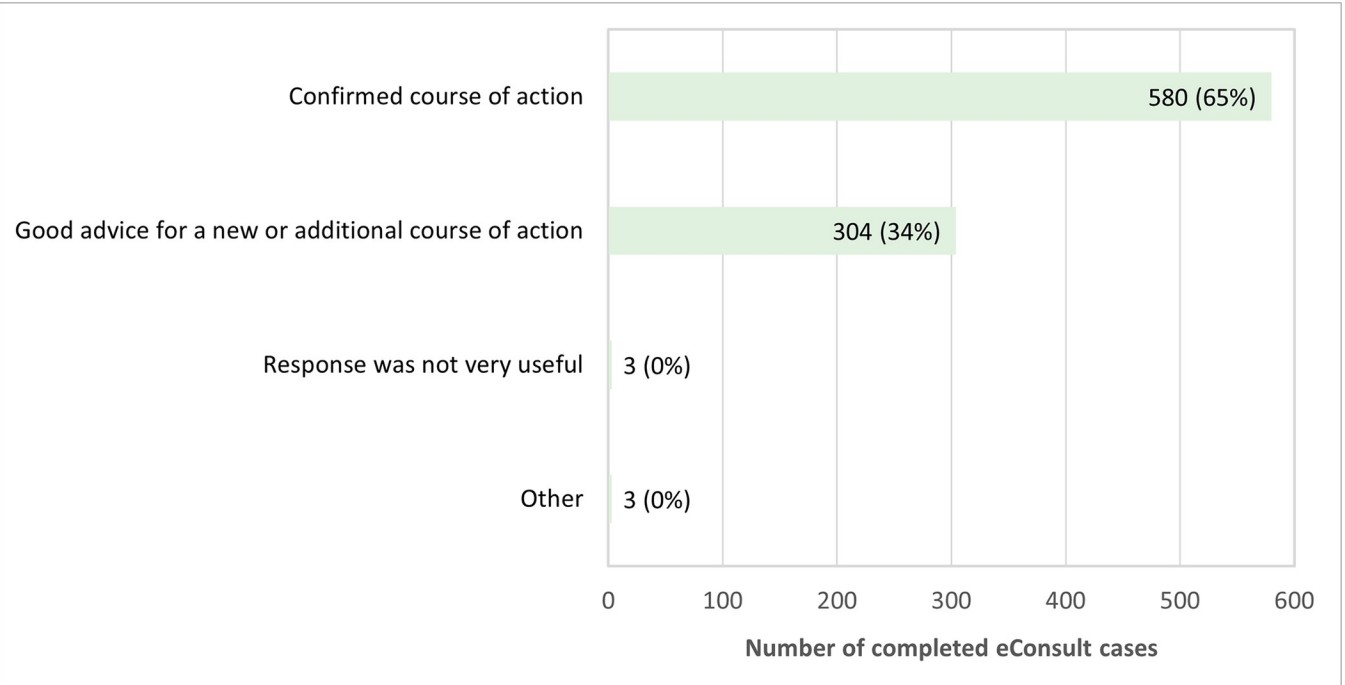

**Fig 3. Outcome of eConsult cases (N = 890).**

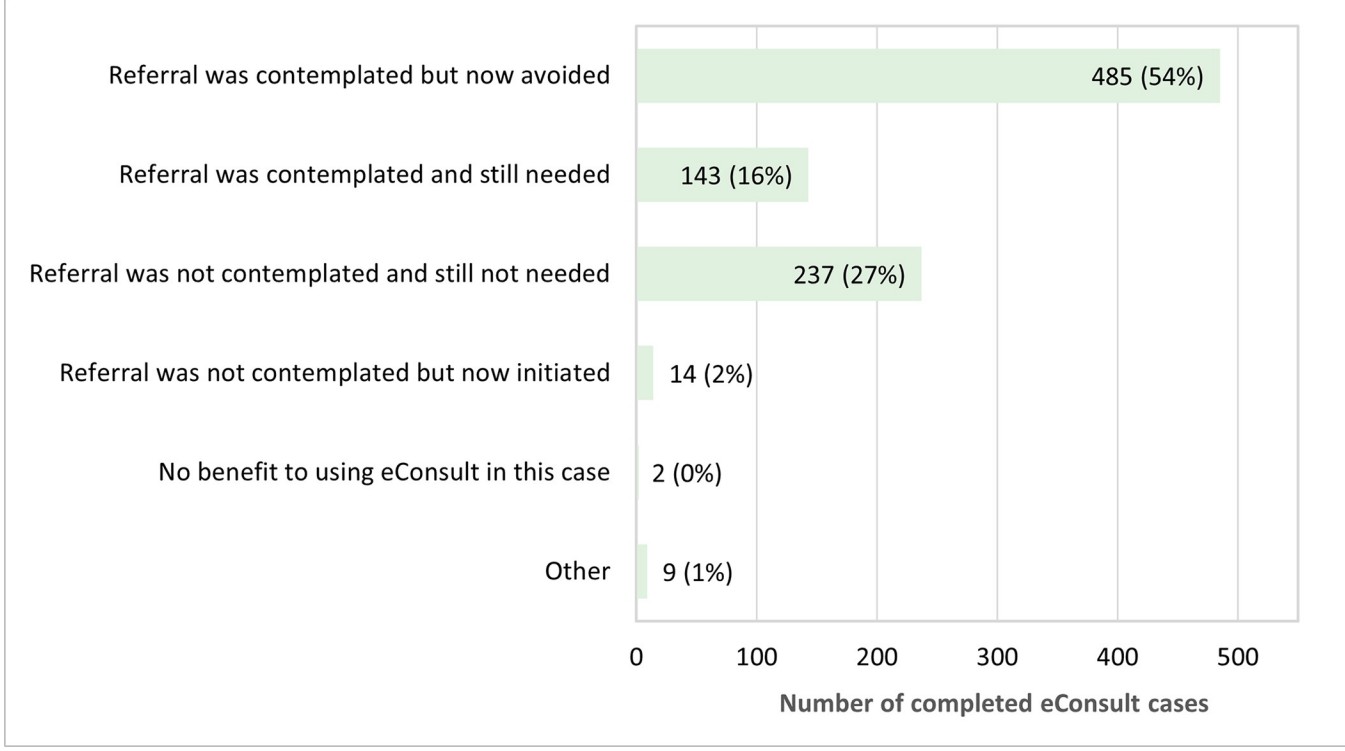

**Fig 4. Referral outcome of eConsult cases (N = 890).**

treatment (OAT) for treating opioid use disorder. This is supported by the Centre of Addiction and Mental Health (CAMH) Opioid Agonist Therapy guidelines, where they recommend conducting a baseline ECG and consulting a cardiologist if a patient undergoing OAT exhibits a prolonged QTc interval [18]. Our hypothesis was also validated by a cardiologist who regularly responds to eConsults, in which they communicated that the predominant nature of eConsults originating from correctional facilities revolves around the adjustment of OAT dosages and calculations related to QT and QTc intervals, and the assessment of other medication lists such as stimulants and antipsychotics to ensure the risk of proarrhythmia remains low (A cardiologist, email communication, July 2023). As of March 2023, the Correctional Service of Canada has reported that 765 incarcerated individuals in federal prisons in Ontario are undergoing OAT, while an additional 11 remain on the waitlist [19]. With a median specialist turnaround time of less than one day, Ontario eConsult allows PCPs to confidently enroll people who are incarcerated in OAT, knowing that they have rapid and timely access to essential cardiology advice. This efficient specialist advice offered through the eConsult platform helps PCPs make well-informed decisions about the implementation and management of OAT for incarcerated individuals, improving care quality and promoting a safer and more effective approach to opioid-related issues in federal prisons.

Poor access to specialist care, decreased patient safety, long wait times, and adverse patient outcomes due to unnecessary transfers and potentially missed referrals are amongst the many previously identified gaps in care within correctional facilities [5, 20]. eConsult not only brings incarcerated individuals the advantage of timely specialist advice, but also presents substantial benefits to the justice system. Approximately 81% of PCPs in our study agreed that a face-to-face referral was not needed following the eConsult, in which 54% had originally intended to refer the patient for an in-person specialist visit prior to the eConsult. Reducing the need for in-person referrals and providing specialist advice through eConsult relieves the department of corrections from escorting the patient, which typically requires at least two prison officers to accompany each incarcerated individual [8]. It also addresses the challenge of waiting for a specialist appointment, especially if the person who is incarcerated is frequently changing institutions. Furthermore, registration and utilization of Ontario eConsult demand minimal training. Start-up costs are virtually non-existent due to the presence of pre-existing technical infrastructure for the service, eliminating the need for additional equipment. This streamlined process mitigates potential security issues and logistical complexities that arise from an in-person specialist appointment, ultimately leading to cost savings for the correctional facilities. In addition, there are similarities between the environments of correctional facilities and long-term care homes, as both encounter obstacles transporting patients outside of their institutions. A previous study demonstrated the efficacy of eConsult in improving access to specialist advice for those living in long-term care [21]. eConsult not only averted unnecessary transfers and reduced costs, but also augmented PCPs' capacity to deliver patient-centered care in accordance with individual care objectives.

To our knowledge, Ontario eConsult is the only service in Canada that allows PCPs in correctional facilities to communicate asynchronously with specialists on behalf of their patients. With this service only available in correctional facilities in Ontario, there is potential to expand eConsult to federal and provincial/territorial correctional facilities in the country's remaining nine provinces and three territories. Of the five countries outside of Canada offering an eConsult solution that serves incarcerated individuals, 511 eConsult requests were initiated in France for 450 adult patients across 8 prison health centers participating in the network of tele-expertise in dermatology between June 2014 and June 2015 [22]. The median response time was 5.0 days, and dermatologists spent less than 6 minutes in their response in 50% of cases. Following the tele-expertise in dermatology for incarcerated individuals, only 2.9% of

patients later required a face-to-face appointment or hospitalization [22]. Tele-expertise was well accepted among physicians with the majority of responders (N = 9/10) willing to continue using it [22]. Additionally, a pilot study carried out in Western Australia assessed the effectiveness of their online eye care system within correctional facilities [23]. The study successfully transmitted data from 11 patients, and the consulting ophthalmologist provided advice within 24 hours [23]. This not only expedited the process but also resulted in a cost savings of A\$440 per medical consultation [23]. Similarly, our study revealed that specialists dedicated a median of 15 minutes to address the eConsults, with a noteworthy median response time of 0.9 days from the moment the eConsult was submitted, underscoring the prompt accessibility of specialist advice for people who are incarcerated in federal custody. The potential expansion of multispecialty eConsult services beyond Canada's borders holds the promise of enhancing the access to healthcare for incarcerated individuals. However, it is important to note that potential challenges for implementation have been identified, including regulatory issues, initial start-up costs, the need for administrative support, and addressing the demands of training and technical intricacies [24–26]. Nevertheless, the prevailing sentiment in the literature remains consistently positive, lending strong support for the use of eConsult in correctional settings.

Lastly, there was a remarkable surge in the utilization of Ontario eConsult by PCPs in correctional facilities during the COVID-19 pandemic in Ontario (Fig 1). The volume of eConsults increased by 59% from the fiscal year 2019–2020 (pre-COVID-19) to 2020–2021 (post-onset of the COVID-19 pandemic). Interestingly, during the same two-year period, there was a simultaneous decrease of 9% in the average counts of incarcerated individuals in federal custody in Ontario [3]. An increase in the utilization of eConsult during the pandemic time frame was also observed outside of correctional facilities in Ontario, where case volumes in the community increased by 71% between February 2020 and November 2020 [10]. The COVID-19 pandemic led to the closure of numerous specialist offices during lockdown, or the restriction of outpatient services exclusively to essential visits. Consequently, there emerged a notable upsurge in the utilization of virtual tools such as eConsult by PCPs seeking specialist advice for their patients, both inside and outside of correctional facilities.

## Limitations

Our eConsult data represents only one segment of the country and may not be fully generalizable to the national or international context. The study relied on valuable utilization data and survey results; however, future studies should incorporate additional data sources such as case logs, patient-level data, or electronic medical record surveys.

## Conclusions

Ontario eConsult has a positive impact in correctional facilities and reduces the need for face-to-face referrals. This valuable tool is cost-effective, improves access to specialist care for people who are incarcerated, and eliminates unnecessary transportation with potential security concerns. Results of this study serve as a template for broader implementation, thereby supporting its uptake by additional correctional facilities in Ontario and across Canada. The results of this study will also aid in the implementation of eConsult in similar institutionalized settings, such as psychiatric hospitals and long-term care homes, where access to care and transportation of patients remains a challenge. Future studies should explore the types of questions being asked by PCPs in correctional facilities and include an economic analysis to evaluate the cost savings and benefits of eConsult in these institutions. This information would

better support decision makers and advocates to implement eConsult in correctional facilities across Ontario and Canada.

## Acknowledgments

The authors wish to thank the PCPs and specialists who use Ontario eConsult, as well as Sheena Guglani for her support in this project.

## Author Contributions

**Conceptualization:** Erin Keely, Clare Liddy.

**Formal analysis:** Danica Goulet, Erin Keely, Clare Liddy.

**Methodology:** Claire Sethuram, Erin Keely, Clare Liddy.

**Project administration:** Erin Keely, Clare Liddy.

**Visualization:** Danica Goulet.

**Writing – original draft:** Danica Goulet.

**Writing – review & editing:** Claire Sethuram, Erin Keely, Clare Liddy.

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
