## [Decision Letter · Decision Letter 0]

6 Oct 2024

PONE-D-24-21042Improving access to specialist care in correctional facilities through Ontario eConsultPLOS ONE

Dear Dr. Liddy,

Thank you for submitting your manuscript to PLOS ONE. After careful consideration, we feel that it has merit but does not fully meet PLOS ONE’s publication criteria as it currently stands. Therefore, we invite you to submit a revised version of the manuscript that addresses the points raised during the review process.

Of particular note, one of the reviewers highlights the importance of using humanizing language for people/patients who are incarcerated. The US Centers for Disease Control and Prevention offer a concise list of recommended language: https://www.cdc.gov/healthcommunication/Preferred_Terms.html. Please pay special attention to review for non-essentializing, humanizing language in addition to addressing the other concerns raised by the reviewers.

We look forward to receiving your revised manuscript.

Kind regards,

Andrea Knittel

Academic Editor

PLOS ONE

“Funding for this project was provided by the Ontario Ministry of Health and Long-Term Care (MOHLTC). The opinions, results, and conclusions reported in this paper are those of the authors and are independent of the funding sources. No endorsement by the Ontario MOHLTC is intended or should be inferred.”

“Funding for this project was provided by the Ontario Ministry of Health and Long-Term Care (MOHLTC). The opinions, results, and conclusions reported in this paper are those of the authors and are independent of the funding sources. No endorsement by the Ontario MOHLTC is intended or should be inferred.”

“Funding for this project was provided by the Ontario Ministry of Health and Long-Term Care (MOHLTC). The opinions, results, and conclusions reported in this paper are those of the authors and are independent of the funding sources. No endorsement by the Ontario MOHLTC is intended or should be inferred.”

“EK is the Executive Director and CL is the Evaluation Lead of the Ontario eConsult Centre of Excellence, which is responsible for the Ontario eConsult program, with both receiving salary support from the Ontario Ministry of Health. EK answers occasional eConsults (less than 1 per month) as a specialist through the service, for which she is reimbursed.”

6. We note that your Data Availability Statement is currently as follows: [All relevant data are within the manuscript and its Supporting Information files.]

Reviewers' comments:

Reviewer's Responses to Questions

**Comments to the Author**

1. Is the manuscript technically sound, and do the data support the conclusions?

Reviewer #1: Yes

Reviewer #2: Yes

2. Has the statistical analysis been performed appropriately and rigorously? 

Reviewer #1: Yes

Reviewer #2: Yes

3. Have the authors made all data underlying the findings in their manuscript fully available?

Reviewer #1: Yes

Reviewer #2: Yes

4. Is the manuscript presented in an intelligible fashion and written in standard English?

Reviewer #1: Yes

Reviewer #2: Yes

5. Review Comments to the Author

Reviewer #1: Thank you for the opportunity to review this paper, which adds to the important body of work demonstrating the value of provider-to-provider eConsult services, in this case for a particularly vulnerable group. I have a few comments the authors may consider for strengthening the work.

Introduction

1. I was curious about the sentence acknowledging poorer health outcomes among offenders and the statement about latent TB, sexually transmitted infections, Hep C, HIV and other blood borne infections, does this sentence capture the poorer health outcomes experienced by offenders? I would have expected poorer physical health, mental health and premature mortality more broadly among the group.

2. Can you provide a couple of examples of the single speciality eConsult services operating in correctional facilities.

Methods

1. It is not clear what the average number of eConsults sent per 1,000 offenders adds to the paper, especially considering the associated limitation outlined in the limitations section. Further clarification and explanation are needed in the methods and results sections about eConsults sent per 1,000 offenders, this would help clarify the statement in the discussion on page 13 that the “The volume of eConsults increased by 59% from the fiscal year 2019-2020 (pre-COVID-19) to 2020-2021…”.

2. I wondered if a table comparing the top 5 or so specialties for this setting with those at the provincial level would be of interest to readers.

Minor typos

On page 7. Two instances where “renumeration” should be “remuneration”.

Reviewer #2: Thank you for allowing me to review this paper. The paper presents descriptive data on the use of eConsults in the Ontario Federal Bureau of Prisons. I appreciate that the authors realized that access to timely specialist care is important.

(1) Although I appreciate that in the world of prisons, "offender" is the term used to describe people, this term has not been common in research literature for at least a decade. I found the overall approach to this paper as set up in the introduction to be related to cost-savings rather than actually giving people who are incarcerated what they deserve.

(2) The actual research aspect of this paper is limited: it is a descriptive study showing before/after uptake of speciality consults. There is no discussion of the systems used to educate clinicians about the EConsult, or evaluations of how the use of econsult different between clinicians or institutions.

(3) There is no discussion about the problems with eConsult for people in jail/prison or their opinions of it. My experience as a clinician specialist going to the jails/prisons is that people who are incarcerated prefer to see someone in person. They feel they get better care. It is more humanizing. I could find NO voice from the perspectives of any one incarcerated as a threat in this whole paper.

(4) The first paragraph of the discussion iterates what was already reported in the introductions. It also takes up over a page and is incredibly dense.

(5) The item "referral avoided" is in the Figure 4. What is "referral avoided"?

(6) I cannot tell if the authors mean "eConsult" as a proprietary item? Does this mean sending a consult request electronically? Telehealth is going on all over the world for people who are in prison. Just one site I found describing this for 2021. https://csgjusticecenter.org/2021/04/12/three-things-to-know-about-implementing-telehealth-in-correctional-facilities/. It feels like the authors of the paper really did not investigate what is going on for telehealth in other setting before writing this paper.

6. PLOS authors have the option to publish the peer review history of their article (what does this mean?). If published, this will include your full peer review and any attached files.

Reviewer #1: No

Reviewer #2: No

---

## [Author Response · Author response to Decision Letter 0]

12 Nov 2024

Editor Comment 1

Of particular note, one of the reviewers highlights the importance of using humanizing language for people/patients who are incarcerated. The US Centers for Disease Control and Prevention offer a concise list of recommended language: https://www.cdc.gov/healthcommunication/Preferred_Terms.html

Response

We have addressed the use of the term 'offender' throughout the manuscript, replacing it with ‘people who are incarcerated’ or 'incarcerated individuals' to align with the recommended non-stigmatizing language for this this population. 

Editor Comment 2

Response

- A rebuttal letter was uploaded and labeled “Response to Reviewers”

- A marked-up copy of the manuscript was uploaded and labeled “Revised Manuscript with Track Changes”

- An unmarked version of the revised manuscript without tracked changes was uploaded and labeled “Manuscript”

Editor Comment 3

Response

The manuscript was revised to meet PLOS ONE’s style requirements.

Editor Comment 4

Please provide additional details regarding participant consent. In the ethics statement in the Methods and online submission information, please ensure that you have specified (1) whether consent was informed and (2) what type you obtained (for instance, written or verbal, and if verbal, how it was documented and witnessed). If your study included minors, state whether you obtained consent from parents or guardians. If the need for consent was waived by the ethics committee, please include this information. If you are reporting a retrospective study of medical records or archived samples, please ensure that you have discussed whether all data were fully anonymized before you accessed them and/or whether the IRB or ethics committee waived the requirement for informed consent. If patients provided informed written consent to have data from their medical records used in research, please include this information.

Response

A statement on the specified consent was added to the methods section. The same text was added to the Ethics Statement field of the submission form. 

Editor Comment 5

In the financial disclosure, please state what role the funders took in the study. If the funders had no role, please state: "The funders had no role in study design, data collection and analysis, decision to publish, or preparation of the manuscript." Please include this amended Role of Funder statement in your cover letter; we will change the online submission form on your behalf.

Response

The role of the funders was added to the Financial Disclosure, and the amended Role of Funder statement was added to our cover letter.

Editor Comment 6

We note that you have provided funding information that is currently declared in your Funding Statement. However, funding information should not appear in the Acknowledgments section or other areas of your manuscript. We will only publish funding information present in the Funding Statement section of the online submission form. Please include your amended statements within your cover letter; we will change the online submission form on your behalf. 

Response

Funding information was removed from the Acknowledgements section and was added to our cover letter. 

Editor Comment 7

In the Competing Interests section, please confirm that this does not alter your adherence to all PLOS ONE policies on sharing data and materials, by including the following statement: "This does not alter our adherence to PLOS ONE policies on sharing data and materials.” Please include your updated Competing Interests statement in your cover letter; we will change the online submission form on your behalf. 

Response

The statement was added to the Competing Interests section and is updated in our cover letter.

Editor Comment 8

Response

A minimal dataset cannot be provided due to the potential presence of identifiable information and because the data is owned by a third-party organization (Ontario Health). Data requests can be made through the following form: https://form.asana.com/?k=zmEBJuRNGb9_YLGOXyqYLg&d=423266023889752 

Reviewer #1 Comment 1

I was curious about the sentence acknowledging poorer health outcomes among offenders and the statement about latent TB, sexually transmitted infections, Hep C, HIV and other blood borne infections, does this sentence capture the poorer health outcomes experienced by offenders? I would have expected poorer physical health, mental health and premature mortality more broadly among the group. 

Response

We have revised the literature in the introduction to better capture the poorer health outcomes experienced by incarcerated individuals. This includes citations that compare their health outcomes—such as physical health, mental health, and premature mortality—with those of the general population. We appreciate your input in helping us provide a more comprehensive understanding of this important issue.

Reviewer #1 Comment 2

Can you provide a couple of examples of the single speciality eConsult services operating in correctional facilities. 

Response

We have included examples of single specialty eConsult services currently operating in correctional facilities in the introduction of the revised manuscript.

Reviewer #1 Comment 3

It is not clear what the average number of eConsults sent per 1,000 offenders adds to the paper, especially considering the associated limitation outlined in the limitations section. Further clarification and explanation are needed in the methods and results sections about eConsults sent per 1,000 offenders, this would help clarify the statement in the discussion on page 13 that the “The volume of eConsults increased by 59% from the fiscal year 2019-2020 (pre-COVID-19) to 2020-2021…”. 

Response

Upon further reflection, we agree that the comparison does not contribute meaningfully to the paper, and we have decided to remove it. Our study focuses solely on volume of eConsults in Ontario’s correctional facilities rather than a comparison with the general public. Therefore, presenting cases per 1,000 is not adding to the paper, especially given the associated limitations.

Reviewer #1 Comment 4

I wondered if a table comparing the top 5 or so specialties for this setting with those at the provincial level would be of interest to readers. The top five specialties accessed by the general public are dermatology (15%), hematology (8%), obstetrics and gynecology (7%), endocrinology and metabolism (7%), and internal medicine (6%). 

Response

We highlighted the only key differences from the general population in the discussion. Specifically, cardiology was the top specialty accessed in the correctional setting, accounting for half of all eConsults, while dermatology, the most frequently accessed specialty by the general public, accounted for 15%. And that cardiology ranks 8th among the general public. 

Reviewer #1 Comment 5

On page 7. Two instances where “renumeration” should be “remuneration”. 

Response

We have corrected both instances of 'renumeration' to 'remuneration' on page 7.

Reviewer #2 Comment 1

Although I appreciate that in the world of prisons, "offender" is the term used to describe people, this term has not been common in research literature for at least a decade. I found the overall approach to this paper as set up in the introduction to be related to cost-savings rather than actually giving people who are incarcerated what they deserve. 

Response

We have addressed the use of the term 'offender' throughout the manuscript, replacing it with ‘people who are incarcerated’ or 'incarcerated individuals' to align with the recommended non-stigmatizing language for this this population. Additionally, we have highlighted the importance of access to care and its impact on improving outcomes for people who are incarcerated in the introduction. 

Reviewer #2 Comment 2

The actual research aspect of this paper is limited: it is a descriptive study showing before/after uptake of speciality consults. There is no discussion of the systems used to educate clinicians about the EConsult, or evaluations of how the use of econsult different between clinicians or institutions. 

Response

The primary objective of this paper was to examine eConsult use within this specific setting (correctional facilities) and population (incarcerated individuals). However, we have published over 100 peer reviewed papers addressing the value of the eConsult service including educational value, its utilization, and its effects on access to specialist care across several Canadian jurisdictions. Additionally, we have also thoroughly explored the types of clinical questions posed through eConsult across various specialties. For a comprehensive list of our publications, please refer to our website: https://www.champlainbaseeconsult.com/publications

Reviewer #2 Comment 3

There is no discussion about the problems with eConsult for people in jail/prison or their opinions of it. My experience as a clinician specialist going to the jails/prisons is that people who are incarcerated prefer to see someone in person. They feel they get better care. It is more humanizing. I could find NO voice from the perspectives of any one incarcerated as a threat in this whole paper. 

Response

Thank you for your insightful comment. This is a great point and direction for future research. The acceptability of eConsults is indeed an important area to explore and would require different methodologies, such as surveys or interviews, followed by qualitative analysis. Understanding the perspectives of incarcerated individuals is crucial for providing patient-centered care and assessing patient experience. While this has not been done in the incarcerated population yet, a relevant study has been conducted by our team with the general population (https://www.champlainbaseeconsult.com/_files/ugd/ac5147_750c12a1d8184014a78f11b059f0d24f.pdf?index=true). Our current paper focuses on eConsult service utilization for people who are incarcerated, and as such focuses on the metrics of service utilization.

Reviewer #2 Comment 4

The first paragraph of the discussion iterates what was already reported in the introductions. It also takes up over a page and is incredibly dense. 

Response

We acknowledge that the first paragraph of the discussion was dense, and we have revised it to enhance clarity and readability. Additionally, we have removed any content that overlapped with the introduction.

Reviewer #2 Comment 5

The item "referral avoided" is in the Figure 4. What is "referral avoided"? 

Response

“Referral avoided” refers to instances where a primary care provider initially considered making a referral but ultimately decided against it due to the guidance provided by the eConsult. To clarify the survey responses, we have revised Fig 4 and expanded on each response in the figure for better understanding.

Reviewer #2 Comment 6

I cannot tell if the authors mean "eConsult" as a proprietary item? Does this mean sending a consult request electronically? Telehealth is going on all over the world for people who are in prison. Just one site I found describing this for 2021. https://csgjusticecenter.org/2021/04/12/three-things-to-know-about-implementing-telehealth-in-correctional-facilities/. It feels like the authors of the paper really did not investigate what is going on for telehealth in other setting before writing this paper. 

Response

Thank you for your inquiry. Some electronic consultation (eConsult) systems across the world are privatized/commercialized, however the Ontario eConsult Service described in this paper is not; it is a provincially available service offered through the Ontario Ministry of Health. It allows primary care providers (PCPs) to submit consult requests electronically to specialists across the province of Ontario. To clarify, eConsult is a type of telehealth that facilitates electronic asynchronous communication between clinicians over a secure, web-based platform. It enables PCPs to communicate and collaborate with specialists to get timely advice on how to manage their patients, often eliminating the need for an in-person visit with the specialist. We have added more detail in the introduction and methods section to better explain the eConsult process and its benefits.

---

## [Editor Report · Decision Letter 1]

18 Nov 2024

Improving access to specialist care in correctional facilities through Ontario eConsult

PONE-D-24-21042R1

Dear Dr. Liddy,

We’re pleased to inform you that your manuscript has been judged scientifically suitable for publication and will be formally accepted for publication once it meets all outstanding technical requirements.

Kind regards,

Andrea K. Knittel, MD PhD

Academic Editor

PLOS ONE

---

## [Editor Report · Acceptance letter]

21 Nov 2024

PONE-D-24-21042R1 

PLOS ONE

Dear Dr. Liddy, 

I'm pleased to inform you that your manuscript has been deemed suitable for publication in PLOS ONE. Congratulations! Your manuscript is now being handed over to our production team.

Kind regards, 

on behalf of

Dr. Andrea K. Knittel 

Academic Editor

PLOS ONE